# Epidermal Growth Factor Receptor Targeting in Colorectal Carcinoma: Antibodies and Patient-Derived Organoids as a Smart Model to Study Therapy Resistance

**DOI:** 10.3390/ijms25137131

**Published:** 2024-06-28

**Authors:** Samuele Tardito, Serena Matis, Maria Raffaella Zocchi, Roberto Benelli, Alessandro Poggi

**Affiliations:** 1Center for Cancer and Immunology Research, Children’s National Hospital, Washington, DC 20010, USA; samuele.tardito@childrensnational.org; 2Molecular Oncology and Angiogenesis Unit, IRRCS Ospedale Policlinico San Martino, 16132 Genoa, Italy; serena.matis@hsanmartino.it; 3Department of Immunology, Transplant and Infectious Diseases, IRCCS Scientific Institute San Raffaele, 20132 Milan, Italy; marazocchi55@gmail.com

**Keywords:** organoids, EGFR, therapeutic antibodies, colorectal cancer, drug resistance

## Abstract

Colorectal cancer (CRC) is the second leading cause of cancer-related death worldwide. Therefore, the need for new therapeutic strategies is still a challenge. Surgery and chemotherapy represent the first-line interventions; nevertheless, the prognosis for metastatic CRC (mCRC) patients remains unacceptable. An important step towards targeted therapy came from the inhibition of the epidermal growth factor receptor (EGFR) pathway, by the anti-EGFR antibody, Cetuximab, or by specific tyrosine kinase inhibitors (TKI). Cetuximab, a mouse–human chimeric monoclonal antibody (mAb), binds to the extracellular domain of EGFR thus impairing EGFR-mediated signaling and reducing cell proliferation. TKI can affect the EGFR biochemical pathway at different steps along the signaling cascade. Apart from Cetuximab, other anti-EGFR mAbs have been developed, such as Panitumumab. Both antibodies have been approved for the treatment of KRAS-NRAS wild type mCRC, alone or in combination with chemotherapy. These antibodies display strong differences in activating the host immune system against CRC, due to their different immunoglobulin isotypes. Although anti-EGFR antibodies are efficient, drug resistance occurs with high frequency. Resistant tumor cell populations can either already be present before therapy or develop later by biochemical adaptations or new genomic mutations in the EGFR pathway. Numerous efforts have been made to improve the efficacy of the anti-EGFR mAbs or to find new agents that are able to block downstream EGFR signaling cascade molecules. Indeed, we examined the importance of analyzing the anti-EGFR antibody–drug conjugates (ADC) developed to overcome resistance and/or stimulate the tumor host’s immunity against CRC growth. Also, patient-derived CRC organoid cultures represent a useful and feasible in vitro model to study tumor behavior and therapy response. Organoids can reflect tumor genetic heterogeneity found in the tissue of origin, representing a unique tool for personalized medicine. Thus, CRC-derived organoid cultures are a smart model for studying the tumor microenvironment and for the preclinical assay of anti-EGFR drugs.

## 1. Introduction

Colorectal cancer represents the second most common cancer in the world, according to the International Agency for Research on Cancer [1,2]. CRC mortality is linked to relapse and metastasis. The big challenge to extending a patient’s life is the treatment of metastatic or unresectable CRC. The options for CRC care include radical surgery, chemotherapy, radiotherapy, targeted therapy, immune checkpoint blockade (ICB), and cancer vaccines. These treatments can be used either alone or in combination [1,2].

Conventional chemotherapy, based on the well-established FOLFOX (5-FluoroUracile + OXaliplatin), XELOX (Capecitabine + OX), or FOLFIRI (5-FU + IRInotecan) drug associations, directly targeting the tumor cells, is mainly used to reduce relapse after surgery of advanced CRC and can lead to the total elimination of tumor cells (curative chemotherapy). On the other hand, neoadjuvant chemo/radiotherapy is used to reduce the tumor mass in locally advanced or metastatic CRC, allowing for or improving the outcome of subsequent surgery [3].

The immunotherapy of lung, kidney, head & neck cancers, and melanoma has shown high efficacy. Immunotherapy can activate the immune system against tumor cells. Thus, immunotherapeutic drugs do not directly target the tumor like chemotherapeutics but elicit a specific antitumor response. While old approaches aimed to push the immune system by cytokine stimulation caused severe side effects, the development of the immune checkpoint inhibitor (ICI) therapy has been able to provide the desired results [4,5].

Adoptive immune cell therapy is another strategy, based on ex vivo (autologous or allogenic) activated anti-tumor effector T cells such as tumor-infiltrating lymphocytes (TILs); these T cells are trained to recognize tumor-associated antigens (TAA), neoantigens or tumor-specific antigens (TSA) through either the T cell receptor (TCR) or engineered chimeric antigen receptors (CAR) [6]. Also, monoclonal antibodies (mAbs) binding specific targets on the tumor cell surface can trigger tumor cell elimination [3]. This antibody-dependent cellular cytotoxicity (ADCC) is mediated by the binding of the FC region of the therapeutic antibody to activating receptors, such as FCγRs, expressed on the effector lymphocytes [7].

EGFRs can be considered the prototype molecule for targeted cancer therapy [8]. Indeed, the EGFR pathway is essential for epithelial cell proliferation, and the mAb Cetuximab can limit CRC cell proliferation in the absence of downstream activating mutations on the same pathway. This antibody has been approved by the Food and Drug Administration (FDA) in 2004 to treat mCRC, and it is still in use to prolong the overall survival (OS) and progression free survival (PFS) of the treated patients, with low toxicity [3]. Thereafter, several mAbs and small molecule inhibitors have been tested to target CRCs and other tumors in different signaling pathways, such as VEGF/VEGFR, IGF/IGFR, HGF/c-MET, Wnt/β-catenin, Notch, Hedgehog, and TGF-β/SMAD. Among these, the frequent driving mutations in the KRAS/BRAF and PI3K/Akt pathways are particularly promising targets in CRCs [3]. Numerous efforts have been made to increase the clinical efficacy of these compounds and minimize cancer resistance to therapy. However, several drugs against these targets have failed in preclinical and clinical studies [9]. Tumor heterogeneity is one of the main obstacles to the identification of an appropriate treatment as it is difficult to represent it both in vitro and in vivo, thus limiting the value of the existing experimental models [10]. The human tumor microenvironment (TME) and immunity is infrequently mirrored in animal models, and the use of severely immunocompromised mice for human tumor cell testing has introduced several experimental biases [11,12]. Thus, the passage from in vitro to in vivo frequently needs complex and costly strategies (i.e., humanized murine models and patient-derived xenografts (PDX)), pushing scientists to develop valid animal-free alternatives. Accordingly, 3D in vitro cultures, such as tumor cells spheroids and patient-derived organoids have been established. These systems, that can accurately reproduce the first avascular phase of a developing tumor, are suitable for the high-throughput testing of new drugs, developing patient-dedicated strategies, and studying drug resistance [13]. Herein, we focus on the EGFR and the complexity of the EGFR signaling cascade to show why this receptor is a key target for therapy. We will also consider the immune mechanisms of a mAb-targeted therapy, analyzing the main processes of resistance. Eventually, we will report and discuss the generation of 3D patient-derived models to study immunotherapy and targeted therapy, focusing our attention on EGFR. The biological significance of these studies includes highlighting the pros and cons of these methods and how they can be used to improve our understanding of the molecular mechanisms mediating drug resistance.

## 2. EGFR Expression and Signaling

A description of the biological role of EGFR will help us to better understand why this surface molecule can be a suitable target for impairing the growth of mucosal epithelial cells.

### EGFR Pathway

EGFRs belong to the ERBB (erythroblastosis oncogene B)/HER (human epidermal growth factor receptor) protein family, comprising four members as follows: HER1 (ERBB1—known also as EGFR), HER2/neu (ERBB2), HER3 (ERBB3) and HER4 (ERBB4) [3,14]. Each member of the ERBB receptor family is composed of four extracellular domains for ligand binding (with the exception of HER2), plus a transmembrane portion, a cytoplasmatic tyrosine kinase (TK) domain, and a carboxy terminal region, which contains tyrosine autophosphorylation sites with regulatory function [15,16].

HER receptors are unique in their mechanism of dimerization. In fact, the ligand does not act as a direct “cross-linker”, as it happens for other TK receptors, but induces a conformational change in the single receptor (linearization of the four extracellular domains), allowing for activation and dimerization [17]. This characteristic is particularly interesting for antibody targeting, as the double-fragment antibody-binding (FAB) region is not able to act as the cross-linker for an active EGFR dimerization.

EGFR can be activated by specific ligands [EGF, transforming growth factor alpha (TGF-α), and amphiregulin (AREG)] or by ligands shared with ERBB4 [betacellulin (BTC), heparin-binding epithelial growth factor (HB-EGF), and epiregulin (EREG)]. The activation of the signaling cascade is triggered when the cytoplasmatic region of the receptor is phosphorylated, favoring the homo- or heterodimerization with HER2, HER3, or HER4. Examples of the main downstream signaling pathways are the RAS/RAF/MEK/ERK, PI3K/AKT, and JAK/STAT3 (Janus kinase/signal transducer and activator of transcription 3) (Figure 1).

Notably, EGFR can interact with other TK receptors such as the hepatocyte growth factor receptor (MET) and the insulin-like growth factor receptor 1 (IGF1R). This interaction can lead to a cross-activation of different signaling pathways affecting the response of epithelial cells to different growth factors [15]. The growth and progression of many cancers, including CRC, could be driven by EGFR. In fact, EGFR can actively control cell proliferation, migration/invasion, differentiation, and resistance to apoptosis.

After dimerization, EGFR is internalized mainly by clathrin-coated pits, although a clathrin-independent pathway is contemporarily activated when high concentrations of ligand are present. Dimerization leads to multiple phosphorylation of the kinases and C-terminal regulatory domains. This culminates in the recruitment of the Src homology 2 (SH2) and phosphotyrosine-binding (PTB)-containing signaling proteins. These proteins can transduce the signal to several signaling routes, which ultimately affect almost all the key functions of the epithelial cell, such as proliferation, differentiation, invasion, migration, survival, and mechanisms of cell repair [18,19].

Hereafter, a brief description of these signaling pathways is provided.

EGFR homodimerization is typically involved in the Ras–Raf–Mek signaling cascade, finally triggering Erk1–Erk2 activation [20,21]. Erk1–Erk2 can then translocate into the nucleus where they phosphorylate several targets, comprising two fundamental transcription factors, Elk1 and C-Myc, to induce cell proliferation [22].

Another important pathway is the phosphatidylinositol 3-kinase (PI3K)–PDK1–Akt–mTOR–6SK pathway, a signaling cascade that informs the cell about the abundance of nutrients, inducing anabolism, protein synthesis, growth, and apoptosis resistance. An EGFR homodimer is unable to directly activate this pathway, but it can be induced by heterodimerization with Her2, Her3, or Her4, or by the downstream target, RAS [23].

The STAT proteins cascade is a pathway particularly involved in tumor progression, oncogenesis, and angiogenesis [24]. It is activated upon receptor dimerization, causing Src recruitment and phosphorylation via the SH2 domains. Once the STAT proteins are phosphorylated by Src, they translocate from cytoplasm into the nucleus, where they enhance the expression of specific genes, including Myc, Nos2, p21, and cytokines [25,26].

Phospholipase C gamma (PLCγ) membrane-associated enzymes are the mediators of another important pathway that could be activated by receptor dimerization [27].

These enzymes participate in ion channel regulation, mediating cell migration and calcium-mediated signaling. EGFR-activated PLCγ interacts with and hydrolyses phosphatidylinositol 4,5-diphosphate (PIP2) to produce inositol 1,3,5-triphosphate (IP3) and 1,2 diacylglycerol (DAG). While IP3 increases intracellular calcium levels, DAG favors the activation of protein kinase C (PKC). The activated PKC, in turn, activates MAPK and c-Jun NH2-terminal kinase [28].

EGFR promotes also the Nck/PAK signal cascade (mediating cell migration and survival) [29]. The Nck adaptor protein contains an SH2 domain mediating the docking on EGFR. Nck, in turn, interacts with and activates PAK. Activated PAK triggers JKNs (c-Jun kinase) through the MEKK1–MKK4/7 cascade, thereby enabling JNK to migrate into the nucleus and phosphorylate transcription factors such as c-Fos and c-Jun [30].

Finally, EGFR is able to cause its own endocytosis in certain conditions. The recruitment of Cbl causes EGFR internalization and ubiquitination. Endosomes containing EGFR can eventually be degraded by a fusion with lysosomes or be recycled to the cell surface. This difference in fate is frequently mediated by the strength of the interaction between EGFR and its ligands, where a strong binding pushes the receptor to degradation [31]. The most frequent EGFR modification in tumor cells is overexpression. This phenomenon might conduct to ligand-independent receptor dimerization or transactivation, triggering non-canonical signaling and resulting in the activation of the transcription factor interferon regulatory factor 3 (IRF3) [32,33]. Upon ligand stimulation (only EGF, HB-EGF, TGF-α, β-Cellulin, and EREG) [34], EGFR can continuously evade lysosomal degradation and then move to nucleus, where it promotes the transcription of target genes, such as Cyclin D1, STAT or/and E2F1 [35]. Nuclear localization of EGFR is related to disease severity thus conferring resistance to therapeutic antibodies with anti-proliferative properties [36].

## 3. EGFR-Targeted Therapy

The upregulation of the EGFR pathway in CRC cells indicates this receptor as a promising therapeutic target, a hypothesis confirmed by the efficacy of the current anti-EGFR inhibitors [8].

EGFR signaling suppression can be achieved by two different strategies as follows: anti-EGFR antibodies (causing EGFR internalization and directly avoiding the binding between ligand and receptor) or small molecule tyrosine kinase inhibitors (preventing EGFR phosphorylation). These could be used either in monotherapy or in combination with chemotherapeutic drugs [37].

### 3.1. Targeting the Extracellular Domain of EGFR

The first therapeutic antibody designed to block EGFR is Cetuximab (IMC-C225, Erbitux ImClone Systems Inc., New York, NY, USA), first produced in 1983. Cetuximab blocks the ligand-binding domain of EGFR, limiting the proliferation of various epithelial tumor cell lines. Cetuximab is a humanized mouse monoclonal antibody composed of the Fv (variable regions) portion of a murine anti-EGFR antibody with human IgG1 heavy and light chain constant regions. The FDA approved Cetuximab in 2004 for the treatment of mCRCs and primary or recurrent head and neck squamous cell carcinoma (HNSCC) [38]. Cetuximab causes the internalization of EGFR to the endoplasmic reticulum (ER) and/or to the cellular nucleus thus blocking EGF from binding to the EGFR [39] without triggering its phosphorylation [40]. Moreover, Cetuximab avoids the EGFR extracellular region from embracing the comprehensive conformation required for dimerization [41]. In addition to the clearance of EGFR from the tumor cell surface, this antibody can indirectly impair tumor growth, inhibiting further angiogenesis, invasion, and metastasis by targeting the tumor stroma. These effects act in synergy with chemotherapy and radiotherapy. Notably, Cetuximab can also induce ADCC, that is mediated by an antibody binding the antigen on a target cell, while its FC portion is recognized by the FCγR expressed on effector cells [42]; however, the outcome of this target–effector bridge depends on the features of the effector cells [42]. Indeed, the FC receptors are expressed on several immune cells involved in innate immunity, such as monocytes, monocyte-derived cells, other myeloid cells, as well as natural killer (NK) cells, and subsets of T lymphocytes [43]. Importantly, Cetuximab can be directly immunogenic in about 5% of treated patients. In fact, the Fv murine-derived region can trigger an immune response leading to the inactivation of Cetuximab itself. To overcome this problem, the full human antibody Panitumumab was developed through the immunization of transgenic mice (XenoMouse) [44,45]. Panitumumab was approved by the U.S. FDA in 2006 as a possible substitute of Cetuximab for the same applications. The discovery of the irresponsiveness of KRAS-mutated mCRCs to the anti-EGFR therapy caused the EMA (in 2009) and FDA (in 2011) to restrict the use of Cetuximab and Panitumumab to chemotherapy-refractory mCRC patients with a wild-type KRAS.

In particular, Cetuximab is used as the first-line treatment of KRAS-WT EGFR mCRCs, either in combination with irinotecan, fluorouracil, and leucovorin, or in patients who are refractory to irinotecan therapy in combination with irinotecan, or as a single agent in patients where irinotecan- and oxaliplatin-based therapies have failed. It is of note that the pattern of KRAS mutations differ among the different cancer types and KRAS mutations are usually represented by single-base missense. The majority of these mutations are found at codon 12 (G12), codon 13 (G13) or codon 61 (Q61) [44,46,47,48,49,50]. It has been recently demonstrated using CRC cell lines and patient-derived organoids that specific inhibitors of the KRASG12C form of KRAS are efficient only when used in association with anti-EGFR antibodies [51]. This finding may suggest that anti-EGFR antibodies can also be used in KRAS-mutated patients in the near future and within specific clinical settings.

The differences between Cetuximab and Panitumumab have been already described in great detail [38] and are summarized in Figure 2.

Briefly, it appears that the two antibodies recognize different epitopes on the ligand-binding domain, although partly overlapping [52]. More importantly, Panitumumab shows a higher dissociation constant (K_D_) than Cetuximab, and both the antibodies show a greater affinity for EGFR than EGF [52,53]. Conceivably, these antibodies could activate different effector immune cells related to the different antibody isotypes. Indeed, the IgG1 Cetuximab can trigger efficient NK cell-mediated ADCC, aside from complement dependent cytotoxicity (CDC), while the IgG2 Panitumumab is not able to trigger this response [38,54] (Figure 3).

The receptors for IgG1 and IgG2 can be expressed on innate immune cells other than NK cells, such as neutrophils and monocytes [38]. Indeed, both mAbs can activate these lymphoid/myeloid populations when present in the TME. It is not clear whether the activation of immune cells is involved in patient response [38]. A recent review looking at the overall survival and progression free survival of patients treated with either one of these mAbs found no evident differences (Figure 4) [38]. On the other hand, an association among the patient response to Cetuximab, efficient NK ADCC-mediated killing in vitro, and FcγR polymorphisms has been found [55].

Overall, it appears that the involvement of the immune system in Panitumumab-treated CRC patients is lower than with Cetuximab. However, no clear advantages or disadvantages have been definitively reported. It is conceivable that the different degrees of FCR activation (in favor of Cetuximab) is balanced by the different degrees of affinity for the EGFR (in favor of Panitumumab), possibly favoring the indirect versus direct effects of the two antibodies. Also, this would suggest that these antibodies could be empowered by the conjugation with cytotoxic drugs or immune-activating molecules to enhance the anti-CRC effect.

### 3.2. Targeting the Intracellular Domain of EGFR

This topic has been well reviewed in other reports and, herein, we provide general information to complete the scenario of the potential inhibitors of EGFR-mediated signaling [56]. Tyrosine kinase inhibitors (TKI) (such as Erlotinib and Gefitinib) can bind to the tyrosine kinase domain of EGFR and impair the activity of this receptor [57].

In fact, they do not alter the surface expression of EGFR but block the EGF-mediated signaling instead [58]. Gefitinib (also called ZD1839, Iressa^®^) is a highly specific TKI for EGFR and has been effective in preclinical models, but in phase II trials, the need to improve patients’ selection has become evident. Gefitinib was found to be effective against mCRCs when associated with chemotherapy [59], suggesting a possible synergy with antiblastic drugs, although this finding could be derived from studies with some detection and selection bias [60,61]. Overall, the use of EGFR TKI against mCRC did not improve overall survival and has since been abandoned [61].

## 4. Mechanism of Resistance to Antibody Therapy

Despite the efficacy of Cetuximab, it has been clear from the beginning that some patients do not respond to EGFR inhibition, and patients with an initial benefit could show frequent relapse [62,63]. Numerous studies have identified the genetic markers and signaling pathways involved in primary or acquired resistance to EGFR targeting [62].

Basically, this resistance can be mediated by the alteration of the extracellular environment (such as EGFR/ligand expression) or the bypass of EGFR for intracellular signaling propagation (Figure 5).

### 4.1. Resistance Mediated by Alterations of EGFR or Its Ligands

Up to 25% of mCRC patients’ resistance to Cetuximab is mediated by mutations affecting the EGFR ectodomain, preventing antibody binding [64]. These mutations quite specifically affect the small portion of domain III recognized by the antibody, mainly the S492, G465, S464, and V441 residues [65]. Importantly, the S492R mutation can affect the binding of Cetuximab, without any effect on Panitumumab [52,66]. Another parameter that could affect patients’ overall response rate is the EGFR gene copy number. Studies by Moroni et al. [67,68] have shown that most non-responder patients have reduced EGFR copy numbers. However, the quantification of EGFR copy number variations is not applied to clinical practice, and its definitive relevance as a determinant of Cetuximab efficacy remains undefined [69].

Intuitively, a parameter affecting the oncogenicity of EGFR is the contemporary presence of its ligands. As the anti-EGFR-inhibiting antibodies compete with the EGFR ligands due to their binding affinities with the receptor, it is conceivable that a low EGFR ligand expression could affect the efficacy of these antibodies [41,70]. Indeed, it has been observed that in vitro cancer cell lines as well as patients with high expression of EGFR ligands, such as amphiregulin (AREG) and epiregulin (EREG), benefit more from Cetuximab therapy [71,72].

Of note, these ligands are usually produced by the cancerous cell, triggering an autocrine activation of the receptor. On the other hand, Cetuximab treatment induces increased systemic levels of EGF in mCRC patients [73] that could compete with the antibody for EGFR binding [74].

### 4.2. Resistance Mediated by Mutations of the EGFR-Associated Signaling Molecules

The RAS protein family is a main mediator of EGFR signaling, and about 50% of the CRCs carry specific activating mutations of these targets [75]. KRAS, HRAS, and NRAS are guanosine-5′-triphosphate (GTP)-binding proteins shared in the signaling of several growth factor receptors, and they act as hubs for the activation of both the Erk and Akt pathways [76,77]. KRAS is the main target of mutations (40% CRC), preferentially hitting codons 12 and 13 in exon 2 [77]. Only 15% of KRAS mutations are detected in exons 3 and 4 [78,79]. These oncogenic mutations lead to the constitutive activation of RAS kinases, bypassing upstream EGF–EGFR signaling and nulling anti-EGFR antibodies efficacy. Consequently, RAS mutations are negative prognostic markers in metastatic CRCs [77].

Another important player in EGFR target therapy resistance and aberrant activation of signaling is BRAF, the downstream effector molecule of RAS [80].

BRAF mutations affect a minority of CRCs (about 10%). In this cohort of CRC patients, the most frequent BRAF mutation (95% frequency) is a single aminoacidic substitution in the V600 residue (V600E being the most represented). BRAF V600E is a constitutively active monomer, inducing a strong signaling in the downstream kinases Mek1–2, stimulating tumor cell proliferation and survival [81].

While BRAF mutations were among the exclusion criteria for Cetuximab therapy in mCRCs, the BEACON trial has shown that the combination of Cetuximab with the BRAF inhibitor, Encorafenib, has therapeutic efficacy. Cetuximab complements the BRAF V600E inhibitor in CRC cells blocking the signaling reactivated by the EGF-dependent wild-type pathway [82].

Also, the activation of the Akt pathway can compensate for the block of EGFR, mainly by phosphatidylinositol-4,5-bisphosphate 3-kinase catalytic subunit alpha (PIK3CA) and phosphatase and tensin homolog (PTEN) target mutations. While PIK3CA is affected by point mutations thus increasing its activity, PTEN (an antagonist of Akt signaling) is usually depleted [83]. The point mutations in PIK3CA can localize in the helical or the kinase domains. Mutations in exon 9 (E542K, E545K) consist of an amino acid substitution of an opposite charge that disrupts the interaction between the regulatory subunit p85α and the catalytic subunit p110α in the helical domain. These mutations confer a gain of function in the binding with RAS–GTP. The mutation of exon 20 (H1047R) produces the constitutive activation of the kinase domain. These mutations represent approximately 80% of the total mutations affecting PIK3CA and are found in 18–20% of CRCs, along with RAS and BRAF mutations [84].

PTEN is a lipid phosphatase acting as a tumor suppressor gene; contrasting PI3K enzymatic activity, it switches off the downstream Akt signaling cascade. The biallelic inactivation of PTEN by silencing or depleting mutations causes an uncontrolled upregulation of Akt signaling, inducing tumor cell proliferation and survival [85,86].

Finally, it is still debated if the excessive activation of the JAK/STAT pathway by increased STAT3 phosphorylation is implicated in anti-EGFR mAb resistance [87].

Indeed, it has been shown that Cetuximab can promote the sensitivity of the CRC cells to the irinotecan metabolite SN38, impairing the expression of Heat Shock Protein 27 and blocking the Jak/STAT signaling pathway [88]. This finding would suggest that STAT3 blocking is also relevant for Cetuximab-based therapy in wild-type RAS CRC. Although it is well known that STAT3 activation is involved in some features of cancer cells, STAT3 inhibitors have not been used to treat CRCs yet; however, there are inhibitors that have shown encouraging effectiveness in preclinical studies that have entered clinical trials [89,90]. Notably, all these mutations can also be generated/selected during the anti-EGFR therapy of originally sensible tumors.

The targeting of EGFR with tyrosine kinase inhibitors can lead to the emergence of resistance and thus limiting the efficacy of the treatment even when several generations of these TKI have been developed. Importantly, the generation of fully resistant cells can pass through a cell state of “drug-tolerant persister” (DTP) [91] that precedes the resister state. In this context, it has been shown that tyrosine receptors such as AXL induce low-fidelity DNA polymerases favoring the generation of resistant cells. This would imply that the upregulation of tyrosine kinase receptors induced by the pharmacological treatment can increase the adaptability of tumor cells by triggering polymerases that are able to synthesize DNA past the damaged bases [92,93,94,95]. In these instances, combination therapies can be a solution to eliminate resistant cells [96].

### 4.3. Other Cell Surface Receptors Can Substitute the EGF–EGFR-Mediated Signaling

The resistance to anti-EGFR targeted therapy can rise also by the selection of tumor clones that use alternative pathways of signaling [97]. Among the growth factor receptors substituting EGFR, there are the type 1 insulin-like growth factor receptor (IGF-1R), the mesenchymal–epithelial transition factor receptor (MET receptor), and the other HER family members.

IGF-1R, when activated by IGF-1 or IGF-2, is able to trigger both the RAS/RAF/MAPK and PI3K/AKT pathways [98]. IGF-1R can not only vicariate EGFR, but also synergize with it, thanks to a strong molecular crosstalk empowering cell proliferation [98,99,100]. MET is a tyrosine kinase receptor, mainly expressed on epithelial and endothelial cells, that binds the Hepatocyte Growth Factor (HGF)/scatter factor (SF) [101]. The HGF is usually produced by mesenchymal stromal cells as an inactive monomer which is stored in the extracellular matrix and activated by different serine proteases linked to the tissue injury response [102]. HGF–Met interactions can mediate downstream signaling, activating the PI3K/AKT, RAC1/cell division control protein 42 (CDC42), RAP1 and RAS/MAPK, and β-Catenin pathways, leading to cell proliferation and favoring survival [103]. Like IGF-1R, the crosstalk between MET and EGFR can lead to an acquired drug resistance to Cetuximab treatment [104,105]. Moreover, MET triggers an invasive phenotype of cells (twist1+) [106], typical of the epithelial-to-mesenchymal transition (EMT), favoring the resistance to anti-EGFR treatment [107,108,109]. Indeed, the cellular shift from an epithelial to a mesenchymal phenotype reduces EGFR involvement in sustaining cell proliferation and survival, while MET activation improves cell motility and the infiltration of the neighboring tissue [110]. The resistance of CRC to Cetuximab or Panitumumab can also be mediated by the activity of other HER family members; for example, by the overexpression of the HER2 gene, an event with quite a low frequency (2%) can occur [14]. Yonesaka et al. reported that the resistance to Cetuximab can also be linked to the production of the HER3/4 ligand Neuregulin/Heregulin. Indeed, HER3 is more expressed in mCRC than in low-stage tumors [111]. Overall, the plasticity of HER2/3/4 in forming active homo/heterodimers can partially compensate the mAbs-induced EGFR blocking in CRC cells, reactivating ERK signaling [112].

The redundancy of HER receptor family members can vicariate EGFR inhibition as the HER2 gene is amplified in 3% of CRC patients and has been linked to a worse prognosis during anti-EGFR therapy [68]. The upregulation of heregulin, the HER3 ligand, has been observed as a primary response to EGFR inhibition [112], contributing to the formation of the HER3/HER3 and HER3/HER2 dimers. Accordingly, the use of a triple targeting of HER receptors has been proposed as a promising approach to target wild-type KRAS CRCs [113]. A recently developed ADC, BCG019, could be a future weapon for this purpose, containing both anti-EGFR + HER3 binding ability and vc-MMAE or BCPT02 payloads [114].

Another escape mechanism of Cetuximab/Panitumumab resistance is the overexpression of AXL, a member of the TAM (TYRO3, AXL, and MERTK) receptor tyrosine kinases family [115]. AXL expression has been associated to a poor prognosis in colorectal cancer patients [116], unrelated to the RAS mutation status. AXL-positive CRC cells show intrinsic resistance to anti-EGFR drugs, and mCRC patients treated with Cetuximab can show increased AXL mRNA levels. In lung cancer, AXL expression was found to be directly linked to EGFR signaling, involving the MAPK and c-Jun pathway activation [117], again suggesting the involvement of HER ligands in AXL upregulation.

An intriguing actor in cancer progression and therapy resistance could also be APOBEC3B, an antiviral DNA cytosine deaminase that contributes to cancer mutation catalyzing cytosine-to-uracil deamination [118]. APOBEC3B has been recently involved in lung cancer resistance to anti-EGFR therapy [119]. EGFR inhibition was caused by APOBEC3B upregulation through NF-kB activation in vitro, and APOBEC3B conferred resistance to the EGFR blockage. The comparison of APOBEC3B levels in NSCLC samples, before or after tyrosine kinase inhibitors treatment, showed an increased expression after treatment, though the median value was comparable, suggesting that this is only one aspect of an overall complicated cellular response to EGFR signaling inhibition.

## 5. Enhancing the Efficacy of Targeting EGFR with Anti-EGFR Antibody–Drug Conjugates

Antibody–drug conjugates (ADC) are composed of a therapeutic antibody linked to one or more molecules of a cytotoxic drug [120]. The main chemical features of ADC have been described in detail elsewhere [121]. Briefly, the antibody acts as a carrier for the drug, to specifically target the desired cell population. The drug can exert its cytotoxic effect only after the endocytosis of the ADC-surface antigen receptor complex [120]. Inside the target cells, the drug is released from the ADC by proteolysis, and it can exert its specific cytotoxic activity. Thanks to Ab specificity, the toxic effect should be confined to tumor cells, sparing normal ones. Notably, the amplification and/or the overexpression of EGFR in tumor cells, compared to normal cells, makes this receptor an optimal target for ADC.

Three anti-EGFR ADC are currently tested for clinical application as follows: ABT-414 (Depatuxizumab Mafodotin), MRG003, and M1231. While ABT-414 and MRG003 are anti-EGFR antibodies linked to an inhibitor of microtubule assembly (monomethyl auristatin F for AT-414 and monomethyl auristatin E for MRG003), M1231 is composed of a bispecific antibody that targets MUC1 and EGFR simultaneously, linked to a hemiasterlin-related payload. The MRG003 is the only ADC that has been tested in a phase 1 clinical trial on CRCs. Although MRG003 showed a manageable safety profile, it did not exert an evident antitumor activity in the CRC patients [122] (Figure 6).

There are also some phase II studies ongoing with MRG003, but those are focused on head and neck cancer [123,124,125,126].

The efficacy of an ADC depends on the level of antigen expressed by the target cells and the cytotoxic effect of the drug conjugate [127]. Accordingly, the mechanisms of resistance could be similar to those reported using the unconjugated antibody. For example, the resistance to the anti-HER2 antibody, Trastuzumab, conjugated to DM1 can be mediated by HER2 downregulation and EGFR compensation [128], or by the altered internalization, lysosomal degradation, and trafficking of the ADC [129,130,131]. The same mechanisms should also be considered for anti-EGFR ADCs [123].

The use of an anti-EGFR ADC linked to a powerful cytotoxic drug should also consider its reactivity with healthy cells. It is of note that the ABT-414 ADC is derived from the native ABT-806 anti-EGFR antibody. This antibody binds to an epitope of EGFR exposed only on tumor cells with an overexpressed or mutated receptor (EGFRvIII), while it is almost inactive on healthy tissues [132] (Figure 6). An improvement in the metabolism of the ADC has been observed in vitro and in animal models using peculiar linkers, such as the triglycyl peptide (CX). CX favors a rapid release of the drug with a stronger antitumor effect, compared to the use of non-cleavable linkers [133,134].

### 5.1. Future Challenges to Improve the Anti-EGFR Targeting Therapeutic Effects

It is essential that an ADC directed against EGFR can reach the target cell and its drug easily be released into the cytoplasm. Bispecific antibodies (antibodies that can recognize two distinct antigens) and biparatopic antibodies (antibodies that recognize two distinct epitopes of the same target molecule) [135] have been studied to increase the rate of internalization and lysosome localization [134]. Using molecular engineering approaches, recombinant antibodies could better penetrate into target cells when tagged with cell penetrating peptides, as well as be directed toward lysosomes with lysosome-sorting peptides [136,137]. These strategies, coupled with cleavable linkers and new drugs with higher and possibly tumor-specific cytotoxic potential, should be designed to maximize the efficiency of intracellular drug release. Also, the development of conjugation methods that ensure a site-specific linking of the drug to the antibody are essential to allow for the production of ADCs with a consistent quality [138,139,140,141,142].

Again, resistance to ADC treatment can arise by several mechanisms reviewed in several recently published reports [143,144,145,146,147,148]. Schematically, any of the steps involved in the targeting of an ADC to a tumor cell can be responsible for the generation of resistance [143,144,145,146,147,148], starting from the recognition of the antigen by the antibody, the features of the payload, altered internalization and lysosomal functions, overexpression of cellular pumps involved in the efflux of drug and the use of alternative intracellular pathways leading to the proliferation of tumor cells.

To plan and increase the efficacy of the anti-EGFR antibody using either native components or ADCs, it is essential to identify whether some resistant clones have come about. In this context, it has recently been found that the mutations involving EGFR-mediated signaling can be detected by the analysis of circulating tumor-derived genetic material present as DNA, tumor cells, or micro-vesicles as exosomes [45,149,150,151,152]. A liquid biopsy can be used to detect resistance mutations against anti-EGFR therapy. However, this analysis is not routinely used to detect the insurgence of resistant tumor cells because the reference cut-off values of the detected mutations are not established, and there are no well-defined protocols to monitor patients. However, with the use of a liquid biopsy, clinicians can follow the evolution and the generation of the acquired resistance in CRC patients. Indeed, the treatment with anti-EGFR antibodies can lead to different mutations in signal transduction receptors such as EGFR, HER2 and MET that are detectable in the circulating blood [151,152]. It is evident that a liquid biopsy can help clinicians in selecting the appropriate therapy but there is not enough evidence to recommend this technique to follow up on patients. However, some ongoing clinical trials will reveal whether the liquid biopsy could be an essential tool to identify resistant cells and consequently modulate therapy (Appendix A) [153,154,155].

Aside from antibodies and ADCs, two main therapeutic tools could, in principle, be applied to CRCs to target the EGFR, namely the engagers and the chimeric antigen receptor (CAR) cells [156,157,158,159,160,161,162,163,164]. Usually, the engagers are bi- or tri-specific antibodies generated by bioengineering that can link with the EGFR on target cells and activate the receptors on lymphocytes (either T or NK cells) [156,157,158,159,160]. This binding evokes the activation of immune cell response, and the proximity between the effector and target cells leads to tumor cell elimination. On the other hand, CAR cells [161,162,163,164] (mainly T or NK) are characterized by a bioengineered receptor with the ability to link with the EGFR, similar to an antibody, and the intrinsic property to activate the immune effector to kill the EGFR-bearing cell. At present, some clinical trials are ongoing using engagers or CAR cells [164], and their clinical use to treat CRC will be possible in the future.

### 5.2. Organoids as an In Vitro Model to Identify and Circumvent the Molecular Mechanisms Mediating Resistance to Therapy

Most research on drug resistance has been based on 2D culture models, relying on half-century-old cell lines adapted to growth on a plastic surface with only the aid of FCS. Not surprisingly, Sato and Clevers’ method to propagate either normal or patient-derived colon organoids (PDO) in 3D cultures has strikingly improved therapy testing, allowing for a patient-specific approach, and identified the key pathways, allowing for normal and cancer stem cell propagation and differentiation [165]. Colorectal organoids, starting from their stem cell component, form spontaneous 3D cell aggregates of enterocytes, and the mucinous and neuroendocrine populations, recapitulating epithelial heterogeneity and polarization/organization. Organoids can be established from mouse or human stem cells, derivable from embryonic, tissue-resident, or induced pluripotent stem cells [166]. Fundamental factors for organoid survival and expansion are beta-catenin agonist signaling, Erk1–2 activation, and BMP–TGFβ–P38 pathway inhibition [167], though only Erk1–2 activation is targeted by anti-EGFR therapy in CRCs. The future direction of organoid testing is pointing towards more complex models, enabling the study of the interactions between cell populations from other lineages, i.e., tumor associated fibroblasts, endothelial cells, macrophages, and lymphocytes [168]. Two mainstream methods are under development, namely the assembloids, attempting to naturally recreate the normal interaction by a direct assembly of different cells, and the organ-on-chip, gating each cell population in a specific area of the chip and allowing for indirect (fluidic) or direct (artificial matrix-controlled) interactions [169].

Keeping these general considerations in mind, locally advanced or metastatic CRC patients show different and frequently unpredictable responses to standard therapy, suggesting the need for a personalized approach. Therefore, this is certainly one of the major goals to improve the therapeutic efficacy of CRC treatment [170]. Large OMIC screenings have been performed in an attempt to identify complex signatures of old and new markers for patient stratification, which has also shown the insufficiency of available therapeutic strategies [171]. Indeed, CRCs can show high heterogeneity, and some specific consensus molecular patterns identifying subtypes of CRC have been defined; however, this categorization is not unique [172]. These subtypes, represented by different genomic alterations and expression profiles, are CMS1 (microsatellite instable [MSI] immune), CMS2 (canonical), CMS3 (metabolic), and CMS4 (mesenchymal). Indeed, only CMS3 has been identified as a new subgroup of CRCs, as the other ones were already defined [173,174,175].

In this context, it is conceivable that PDOs could be a valuable alternative/complement to the complex OMIC definition of each patient [176]. PDOs are patient-specific and could represent a simplified preclinical model for personalized therapy and the identification of intrinsic/acquired drug resistance in advance [177]. Several studies [178,179,180,181] have tested the use of CRC PDOs to identify the responders to standard chemo/radiotherapy. Pasch et al. [182] showed that primary PDOs could be established from 79% of CRC samples and that 49% of PDOs could be passaged at least two times. The match between the tumor of origin and early passage PDOs was high, though some mutations were found in the PDOs and not in the original tumor, suggesting a possible positive enrichment of rare populations in vitro. This finding is not necessarily negative, as these PDOs could anticipate the testing of resident resistant populations unidentified by OMIC-only approaches. These PDOs were tested in a mixed 5-fluorouracil (5FU) + radio therapy, showing how different organoid populations from the same patient could show different sensitivities to treatment. Vlachogiannis et al. [183] prepared a biobank of metastatic gastrointestinal cancers with a 70% establishment success. As in the previous study, these authors noticed that PDOs could be selectively enriched in cell populations with rare mutations, and that would have been lost due to the cut-off of next-generation-sequencing quality controls. PDOs showed a 96% mutational overlap with the biopsy of origin, though a specific enrichment of PDOs with SRC and EGFR amplifications was observed. A good correlation among driving mutations, specific drug targeting, and patients’ response was observed for the available cases (88% positive and 100% negative predictive value). Ooft et al. [184] established, with success, 63% of mCRC PDO, with 56% PDOs available for multiple drug testing. PDOs were treated with 5FU, either in combination with oxaliplatin or irinotecan or with irinotecan alone. PDOs correctly classified 80% and 83.3% of the patients, respectively, that responded to irinotecan alone or associated to 5FU. On the contrary, PDO could not predict the response to 5FU plus oxaliplatin. The authors suggested that this discrepancy could be linked to the contribution of tumor stroma/immune cells (lacking in PDOs) to the overall effect in patients. Wang et al. [185] obtained an overall 77% establishment of primary mCRC cultures. The authors prepared a first cohort of PDO to identify the IC50 of standard chemotherapy combinations (XELOX/FOLFOX or FOLFIRI) that was applied to a second cohort with predictive purposes. Finally, 45 patients were eligible for comparison with their matched PDOs, showing 63% sensitivity, 94% specificity, and 79% accuracy in predicting responses.

These studies have indicated PDOs as reliable models for predicting patients’ response to standard therapy. Accordingly, they also represent a unique tool to test new drug combinations for personalized medicine. Ramzy et al. [186] published an interesting approach integrating PDOs and statistical modeling. This study identified a four-drug, low-dose combination (Regorafenib, Vemurafenib, Lapatinib, and Palbociclib), that outperformed standard FOLFOXIRI treatment in 3D CRC cultures. While this short-term test (72 h treatment) cannot foresee the overall effect in the long term, it represents a good approach for a fast-predictive assay after surgery. Noticeably, the four-drug combination described by these authors, as with broad-spectrum activity, contained two drugs (Regorafenib and Vemurafenib) that were expected to target only specific cells (endothelial cells and BRAF-V600E mutated cells). This implies that some off-target biochemical effects of synthetic inhibitors could turn out therapeutic. Indeed, we recently showed that the P38 inhibitor, SB202190, can mimic the BRAF V600E inhibitor, Dabrafenib, outperforming it in long-term inhibition tests on BRAF-mutated PDOs [187].

PDOs are not only useful for direct drug testing, but also to study tumor–immune cell interactions. We have recently shown that a Cetuximab (Cet)–Zoledronate (ZA) antibody–drug conjugate (Cet–ZA ADC) can trigger Vδ2 T lymphocytes activation, causing expansion and cytotoxicity against CRC PDOs [188] (Figure 7). Notably, the same ADC also triggered the response of Vδ2 T cells against CRC fibroblasts, indicating a promising tool which is able to dually target cancer and its stroma.

Also, the identification of rare genetic profiles by exome and transcriptome analyses can be identified using a three-dimensional (3D) culture system such as mucosa organoids of early-onset CRC. This approach together with the analysis of patient-derived biopsies can also identify tumor suppressor genes and pre-malignant mutations [189,190,191,192]. Altogether, these findings strongly support that the organoids of primary and/or metastatic lesions of CRC can be considered an optimal tool to study the possibility of insurgence of drug resistance [193]. Furthermore, these in vitro micro-tumors can be a good target to study immune reactivity and select drugs that can trigger effector cell-mediated killing of CRC tumor cells [194].

## 6. Conclusions

Several reports have highlighted that the use of anti-EGFR antibodies and/or patient-derived organoid cultures can be considered a useful tool to study the molecular mechanisms involved in therapy resistance in CRC [195,196,197,198,199,200,201]. Despite the efficiency of anti-tumor therapy, it is evident that tumor cells bear a strong plasticity in adaptation to a hostile microenvironment, and a new combinations of treatment should be employed to limit tumor cell growth [202,203,204,205,206]. Indeed, stromal cells and cancer stem cells represent two players that greatly influence the efficacy of therapy [202,203,205]. The resistance to antibody therapy can be overcome using the combination of drugs targeting different biochemical pathways and surface receptors [206,207,208,209]. The activation of the anti-tumor immune response can increase the efficiency of this approach providing strong evidence that hitting several targets at the same time can reduce the selection of resistant tumor cell clones. Organoids of the epithelial cells and assembloids of the epithelia and mesenchymal cells are the present and the future 3D models to assess the mechanisms of resistance [210,211]. Several ongoing studies (26 studies found on ClinicalTrials.gov, keywords: CRC, colorectal cancer, organoids) are using organoids to better define a tailored therapy for CRC patients. A large part of these studies involves recruiting patients. Based on their results, we will understand how much these micro-tumors will help our knowledge and identification of tumor resistance therapy molecular mechanisms. Further, it is conceivable that this approach will allow us to shape the treatment of CRCs.

## Figures and Tables

**Figure 1 ijms-25-07131-f001:**
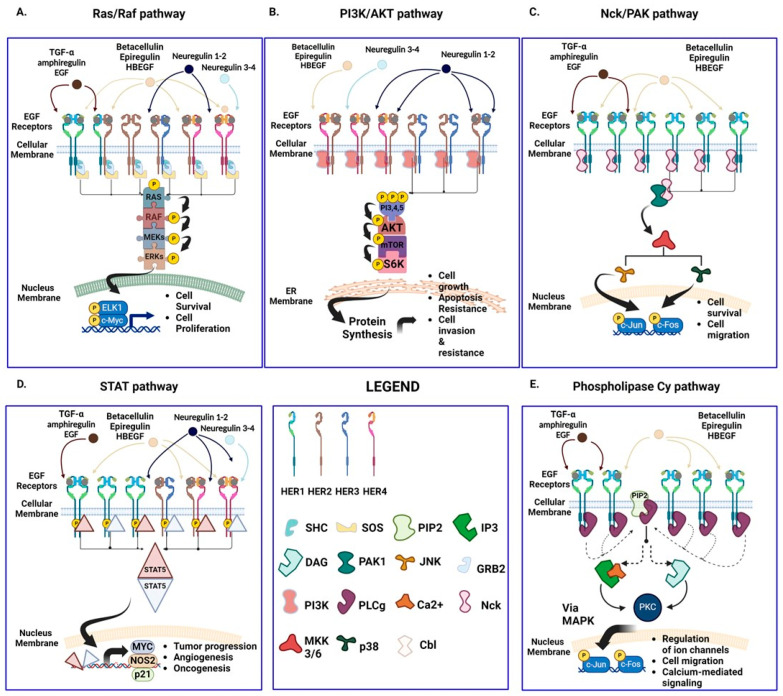
EGFR signaling. The most important biochemical signaling pathways involved upon the engagement of the EGFR. The binding of specific ligands (e.g., EGF, HBEGF, TGF-α, amphiregulin, betacellulin, and neuregulin) can produce up to 10 forms of homo- or heterodimeric receptors of the HER family arising from the intracellular kinase domain conformational modification, which results in autophosphorylation and activation. EGFR signaling involves several pathways including the following: (**A**). Ras/Raf; (**B**). PI3K/AKT; (**C**). Nck/PAK; (**D**). STAT5; and (**E**). Phospholipase Cγ. All these signals may concur to trigger cell proliferation, cell survival, resistance to apoptosis, matrix invasion, and migration of tumor cells. For each pathway, some transducers are shown to point out evidence of the pleiotropic effect induced. SHC: Src Homology domain; SOS: Son of Sevenless; PIP2: phosphoinositol phosphate 2; IP3: inositol-phosphate 3; DAG: Diacylglycerol; PAK: p21-activated protein kinase; JNK: c-Jun N-terminal kinase; GRB2: Growth factor receptor-bound protein 2; PI3K: Phospho-inositol 3 Kinase; PLCγ: Phospholipase C gamma; NCK: non-catalytic region of tyrosine kinase adaptor protein 1; MKK3/6: Mitogen-activated protein kinase–kinase 3/6; p38: protein 38; CBL: Cobalamin. This scheme is not exhaustive of the knowledge of the mechanisms of EGFR-mediated signaling.

**Figure 2 ijms-25-07131-f002:**
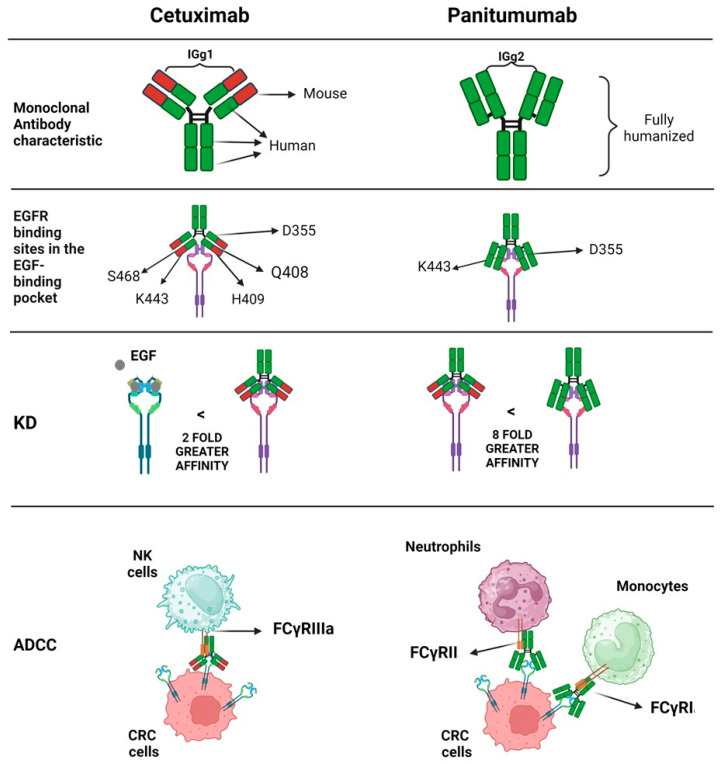
Comparisons between Cetuximab and Panitumumab anti-EGFR therapeutic antibodies. The two therapeutic monoclonal antibodies are markedly different. Indeed, Cetuximab is not fully humanized and can interact with different EGFR-binding sites compared to Panitumumab. Also, this latter antibody shows a stronger affinity for the EGFR (about 8-fold greater). Finally, Cetuximab can trigger antibody-dependent cellular cytotoxicity (ADCC) mediated by the FCγ receptor expressed on NK cells, while Panitumumab can activate other innate cells such as monocytes and neutrophils, as well as their counterparts localized within tissues.

**Figure 3 ijms-25-07131-f003:**
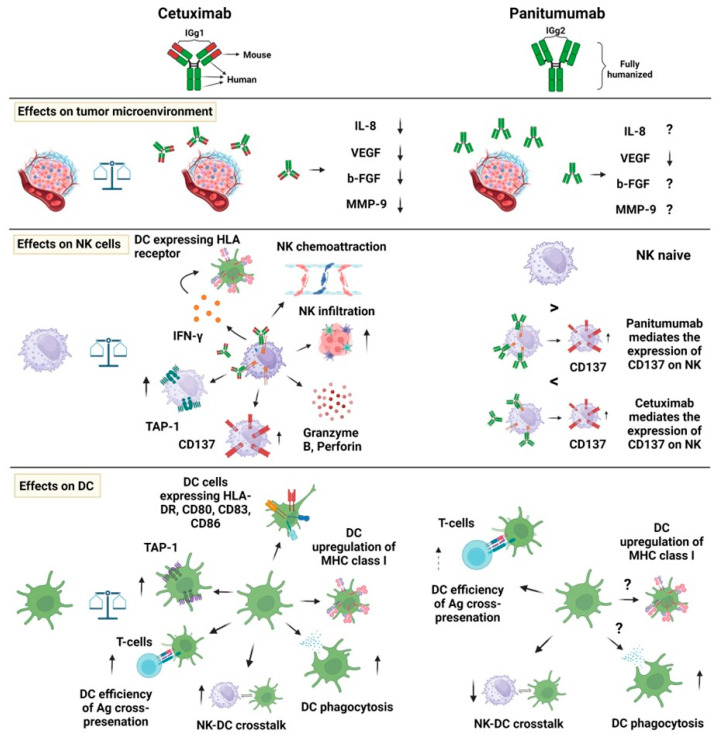
Therapeutic anti-EGFR antibodies Cetuximab and Panitumumab: Differences in immune activation. Some of the main functional effects mediated by the two anti-EGFR therapeutic antibodies are shown. These effects can involve the tumor microenvironment (TME) (upper panel) NK cells (middle panels) and dendritic cells (lower panels). The functional properties of each antibody are mainly linked to the isotype of the immunoglobulin and based on the expression of the appropriate receptors on the effector cells for a given immunoglobulin isotype. The influence on the TME is due to the decrease in the inflammatory cytokines, some metalloproteinases, and growth factors involved in the angiogenesis. On the other hand, the NK cells are activated, leading to the increased expression of CD137 (4-1BB) and secretion of pro-inflammatory factors such as IFN-γ, favoring NK cell homing and infiltration. The NK–DC crosstalk is subsequently induced, leading to the upregulation of molecules involved in the antigen presentation. Several of these effects have been shown for Cetuximab but not for Panitumumab. Legend: MMP: metalloproteinase, VEGF: vascular endothelial growth factor, b-FGF: basic fibroblast growth factor, DC: dendritic cells, NK: natural killer cells, TAP-1: Transporter associated with antigen processing 1, Ag: antigen. The arrows indicate the increase or decrease in a specific function or factors or the possible interactions between the immune cells. ?: it shows that that effect has not been demonstrated but it is conceivable.

**Figure 4 ijms-25-07131-f004:**
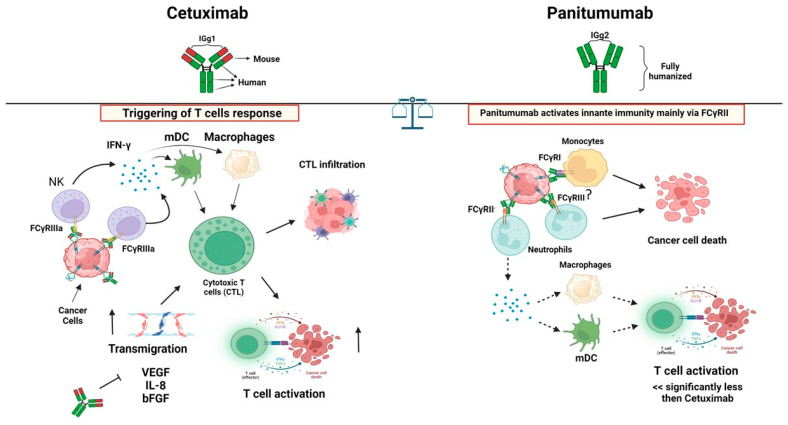
Immune activation mediated by anti-EGFR antibodies. Left panel: The cetuximab-mediated activation of NK cells through the engagement of the FC with the FCγRIII and linkage of EGFR can lead to the production of pro-inflammatory cytokines such as IFN-γ, which, in turn, can trigger the upregulation of molecules involved in the antigen presentation to T cells and their consequent activation. This may induce the localization of effector cells within the tumor site. Right panel: On the other side, the engagement of FCγRI or FCγRII (but not FCγRIII) on the innate cells by Panitumumab can lead to the killing of the tumor cells as well as the induction of maturation of DC. This, in turn, can trigger T cell-mediated tumor cell recognition. It is of note that the effects due to Panitumumab can be significantly weaker than those mediated by Cetuximab. The arrows indicate the increase or decrease in a specific function or factors or the possible interactions and consequences between the immune cells. ?: it shows that that effect has not been demonstrated but it is conceivable.

**Figure 5 ijms-25-07131-f005:**
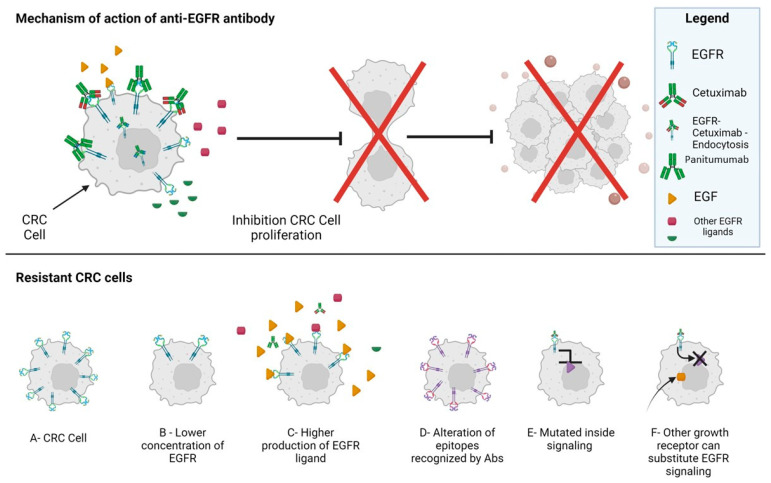
Mechanisms of resistance to antibody-mediated CRC therapy. Upper panel: The two monoclonal antibodies have been developed to block the engagement of the EGFR by its natural ligands. This blockade does not allow signal transduction through the EGFR, leading to reduced cancer cell growth. However, the engagement of the EGFR can also lead to its endocytosis thus reducing the surface expression of EGFR. The anti-EGFR antibodies are efficient only when the downstream EGFR signal transduction is conserved (wild-type EGFR signaling) as in healthy epithelial cells; otherwise, the cancer cells do not need the EGFR-mediated signaling for proliferation. In the lower panel, the main mechanisms of resistance to antibody therapy are shown. In particular, while the original cancer cells express EGFR (A), the resistant cancer cells (either selected upon therapy or with a basal low EGFR expression) reduce the EGFR expression (B), produce higher amounts of EGFR ligands (C), express modified epitopes of EGFR (D), or show the mutation of downstream EGFR signaling (E), or other surface receptors substitute the EGFR thus triggering tumor cell proliferation (e.g., MET/HER2) (F). These mechanisms of resistance can be present in the starting CRC cell population due either to the strong molecular heterogeneity present inside a tumor or selection by the antibody therapy itself. Also, both these molecular mechanisms can be present.

**Figure 6 ijms-25-07131-f006:**
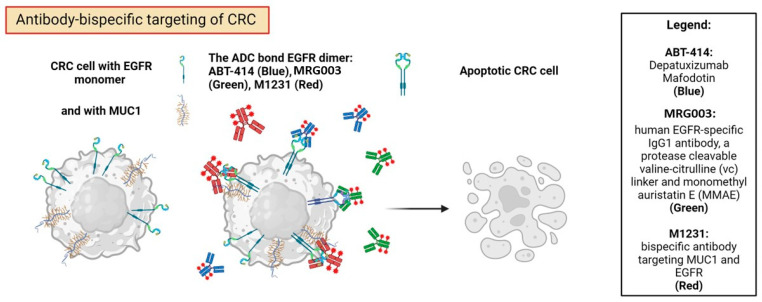
Bispecific antibodies and anti-EGFR antibody–drug conjugates (ADC) to enhance the antitumor effect and limit resistance to therapy. The CRC cells can be targeted with a bispecific antibody that recognizes both EFGR and MUC1 (left). This combination can bypass the resistance due to the reduction in EGFR but it may induce the selection of EFGR-negative and MUC1-negative CRC cells. Also, a potent cytotoxic drug, usually microtubule-targeting drugs such as monomethyl auristatin F or E, can be covalently linked to an anti-EGFR antibody, generating an antibody–drug conjugate (ADC) (middle). This ADC penetrating, after the interaction with EGFR, into the tumor cell can block cell proliferation very efficiently and thus lead to tumor cell apoptosis (right). The use of the antibody as a carrier of the drug defines the specificity of the targeting. This is essential as the drugs used cannot be administered as they stand because of their potent off-target effect. This approach can be applied with low side effects when CRC cells overexpress EGFR compared to healthy cells. The strong antitumor effect limits the generation and negative selection of low-EGFR-expressing CRC cells.

**Figure 7 ijms-25-07131-f007:**
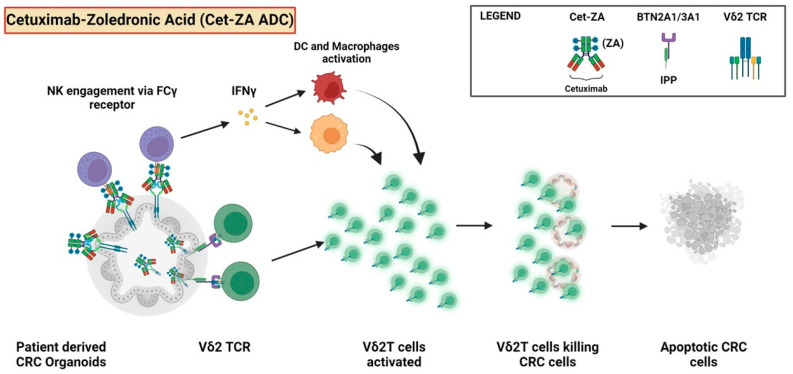
Aminobisphosphonates linked to anti-EGFR antibody cetuximab can trigger the activation of antitumor lymphocytes. Cetuximab (Cet)–Zoledronate (ZA) antibody–drug conjugate (Cet-ZA ADC) can deliver the aminobisphosphonate ZA to tumor cells. The ADC can trigger the activation of FCγRIIIa+ Vδ2T lymphocytes (as well as NK cells). Also, the ZA entering the tumor cells can inhibit the enzymes involved in the cholesterol synthesis, leading to an intracellular increase in small pyrophosphate antigens such as isopentenylpyrophosphate (IPP). In turn, the IPP can be presented at the cell surface through the butyrophilin family members (such as BTN2A1 and BTN3A1) to the T cell receptor of Vδ2T cells. This leads to Vδ2T cell growth and activation of cytotoxic response against autologous CRC PDOs [139]. The arrows indicate the possible steps consequent to the interactions between tumor organoids and the immune cells.

## Data Availability

Not applicable.

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
