# Peer review of "Epidermal Growth Factor Receptor Targeting in Colorectal Carcinoma: Antibodies and Patient-Derived Organoids as a Smart Model to Study Therapy Resistance"

_ijms, 2024, doi:10.3390/ijms25137131_

Round 1

Reviewer 1 Report

Comments and Suggestions for Authors

Colorectal cancer (CRC) is the second leading cause of cancer-related deaths worldwide, necessitating the development of new therapeutic strategies. Surgery and chemotherapy are the primary treatments, but the prognosis for metastatic CRC (mCRC) remains poor. Targeted therapies such as the anti-EGFR antibody Cetuximab and tyrosine kinase inhibitors (TKIs) have shown promise by inhibiting the EGFR pathway, thus reducing cell proliferation. Cetuximab, a chimeric monoclonal antibody, binds to EGFR's extracellular domain, impairing its signaling. Despite the efficacy of anti-EGFR antibodies like Cetuximab and Panitumumab, resistance frequently occurs due to pre-existing or acquired mutations in the EGFR pathway. Efforts to improve anti-EGFR therapies include developing antibody-drug conjugates (ADCs) and using patient-derived CRC organoid cultures for personalized medicine. These organoids mirror the genetic heterogeneity of the original tumors, making them valuable for studying tumor behavior and testing new treatments. This review highlights the significance of anti-EGFR therapies and the innovative approaches to overcome resistance and enhance treatment efficacy for CRC. 

Major Comments: 

1. Authors have to discuss about the classical anti-EGFR therapy and their role in regulation of low fidelity polymerases. 

2.  Authors have to discuss about the compensatory mechanisms mediated by inhibition of EGFR. These include the family of APOBECs, AXL, MERTK HER2 and HER3.  

3. Authors have to include the possible mechanisms of resistance to ADCs

Author Response

Major Comments: 

  • Authors have to discuss about the classical anti-EGFR therapy and their role in regulation of low fidelity polymerases. 
  • Reply Lines 426-438

The targeting of EGFR with tyrosine kinase inhibitors can lead to the emergence of resistance limiting the efficacy of the treatment even when several generations of these TKI have been developed. Importantly, the generation of fully resistant cells can pass through a cell state of “drug-tolerant persister” (DTPs) [91] that precedes the resister state. In this context, it has been shown that tyrosine receptor such as AXL induces low-fidelity DNA polymerases favoring the generation of resistant cells. This would imply that the upregulation of tyrosine kinase receptors induced by the pharmacological treatment can increase the adaptability of tumor cells by triggering polymerases able to synthesize DNA past damaged bases [92–95]. In these instances, the combination therapies can be a solution to eliminate resistant cells [96].

  • Authors have to discuss about the compensatory mechanisms mediated by inhibition of EGFR. These include the family of APOBECs, AXL, MERTK HER2 and HER3.  
  • Reply Lines 471-496

The redundancy of HER receptor family members can vicariate EGFR inhibition: HER2 gene is amplified in 3% of CRC patients and has been linked to a worse prognosis during anti-EGFR therapy [68]. The upregulation of heregulin, the HER3 ligand, has been observed as a primary response to EGFR inhibition [112], contributing to the formation of HER3/HER3 and HER3/HER2 dimers. Accordingly, the use of a triple targeting of HER receptors has been proposed as a promising approach to target KRAS wild type CRC [113]. A recently developed ADC, BCG019, could be a future weapon for this purpose, containing both anti-EGFR+HER3 binding ability and vc-MMAE or BCPT02 payloads [114].

Another escape mechanism of Cetuximab/Panitumumab resistance is the overexpression of AXL, a member of the TAM (TYRO3, AXL, and MERTK) receptor tyrosine kinases family [115]. AXL expression was associated to a poor prognosis in colorectal cancer patients [116], not related to RAS mutation status. AXL-positive CRC cells show intrinsic resistance to anti-EGFR drugs, and mCRC patients treated with Cetuximab can show increased AXL mRNA levels. In lung cancer, AXL expression was found directly linked to EGFR signaling, involving the MAPK and c-Jun pathway activation [117], suggesting again the involvement of HER ligands in AXL upregulation.

An intriguing actor in cancer progression and therapy-resistance could also be APOBEC3B, an antiviral DNA cytosine deaminase that contributes to cancer mutation catalyzing cytosine-to-uracil deamination [118]. APOBEC3B has been recently involved in lung cancer resistance to anti EGFR therapy [119]. EGFR-inhibition caused APOBEC3B upregulation by NF-kB activation in vitro, and APOBEC3B conferred resistance to EGFR block. The comparison of APOBEC3B levels in NSCLC samples, before or after tyrosine kinase inhibitors treatment, showed an increased expression after treatment, though the median value was comparable, suggesting this is only one among an overall complicated cellular response to EGFR signaling inhibition.

In this intricate scenario, the targeting of EGFR with tyrosine kinase inhibitors can lead to the emergence of resistance limiting the efficacy of the treatment even when several generations of these TKI have been developed. Importantly, the generation of fully resistant cells can pass through a cell state of drug tolerant persister [92] that precedes the resister state. In this context, it has been shown that tyrosine receptor such as AXL induces low-fidelity DNA polymerases [93–96] favoring the generation of resistant cells. This would imply that the upregulation of tyrosine kinase receptors induced by the pharmacological treatment can increase the adaptability of tumor cells by triggering polymerases able to synthetize DNA past damaged bases [94,95]. In these instances, the combination therapies can be a solution to eliminate resistant cells [96].

  • Authors have to include the possible mechanisms of resistance to ADCs

Besides some of these mechanisms are shared with native antibody (see figure 5), we have further clarified the mechanisms by which ADC resistance can be generated.

  • Reply Lines 563-586

Again, resistance to ADC treatment can arise by several mechanisms reviewed in several recent published reports [143–148]. Schematically, any of the steps involved in the targeting of an ADC to a tumor cell can be responsible for the generation of resistance [143–148]. Indeed, starting from the recognition of the antigen by the antibody, the features of the payload, altered internalization and lysosomal functions, overexpression of cellular pumps involved in the efflux of drug and use of alternative intracellular pathways leading to proliferation of tumor cells.

Reviewer 2 Report

Comments and Suggestions for Authors

The manuscript focused on Epidermal Growth Factor Receptor targeting in colorectal carcinoma: antibodies and patient-derived organoids as a smart 3 model to study therapy resistance represents a technically correct and timely relevant manuscript available for the publication on this journal.

- In the manuscript, please, could the authors better define the clinical setting where mAB against extracellualr domain of EGFR may be applied in clinical practice? As regards, I would kindly reccomend discussing the molecular landscape of KRAS alteratiosn excluding CRC patients from target therapy.

- In the manuscript, please, could the authors implement some technical aspects on in vitro models to study therapy resistance.

- In this setting, liquid biopsy also emerged as potential tool to guide clinical decision making in resistance setting? Please, could the authors discuss about these approaches?

- In the manuscript, please, could the authors also list the main trials investigating this setting?

Comments on the Quality of English Language

Minor english editing

Author Response

  • In the manuscript, please, could the authors better define the clinical setting where mAb against extracellular domain of EGFR may be applied in clinical practice? As regards, I would kindly recommend discussing the molecular landscape of KRAS alterations excluding CRC patients from target therapy.
  • Reply Lines 249-260

In particular, cetuximab is used as first-line treatment of KRAS WT EGFR mCRC either in combination with irinotecan, fluorouracil and leucovorin or in patients who are refractory to irinotecan therapy in combination with irinotecan or as a single agent in patients where irinotecan and oxaliplatin based therapy failed. It is of note that the pattern of KRAS mutations differ among the different cancer types and usually KRAS mutations are represented by single base-missense. The majority of these mutations are found at codon 12 (G12), codon 13 (G13) or codon 61 (Q61) [44,46–50]. It has been recently demonstrated using CRC cell lines and patient-derived organoids that specific inhibitors of KRASG12C form of KRAS are efficient only when used in association with anti-EGFR antibodies [51]. This finding may suggest that the use of anti-EGFR antibodies can be used also in KRAS-mutated patients in a near future and specific clinical settings. 

  • In the manuscript, please, could the authors implement some technical aspects on in vitro models to study therapy resistance.
  • Reply Lines 602-621

Most research on drug resistance was based 2D culture models, relying of half-century old cell lines adapted to growth on a plastic surface with the only aid of FCS. Not surprisingly, the Sato and Clevers method to propagate either normal or patient-derived colon organoids (PDO) in 3D cultures strikingly improved therapy testing allowing for a patient-specific approach and identifying the key pathways allowing normal and cancer stem cell propagation and differentiation [165].

Colorectal organoids, starting from their stem cell component, form spontaneous 3D cell aggregates of enterocytes, mucinous and neuroendocrine population, recapitulating epithelial heterogeneity and polarization/organization. Organoids can be established from mouse or human stem cells, derivable form embryonic, tissue-resident, or induced pluripotent stem cells [166]. Fundamental factors for organoids survival and expansion are a beta-catenin agonist signaling, Erk1-2 activation, and BMP-TGFβ-P38 pathways inhibition [167], though only Erk1-2 are targeted by anti-EGFR therapy in CRC. The future direction of organoid testing is pointing towards more complex models, enabling to study the interaction of cell population from other lineages, i.e. tumor associated fibroblasts, endothelial cells, macrophages and lymphocytes [168]. Two mainstream methods are under development: the assembloids, attempting to naturally recreate the normal interaction by a direct assembly of different cells, and the organ on chip, gating each cell population in a specific area of the chip allowing an indirect (fluidic) or direct (artificial matrix-controlled) interaction [169].

  • In this setting, liquid biopsy also emerged as potential tool to guide clinical decision making in resistance setting? Please, could the authors discuss about these approaches?
  • Reply Lines 570-587

To plan and increase the efficacy of the anti-EGFR antibody using either native or ADC, it is essential to identify whether some resistant clones are coming out. In this context, it has been recently found that the detection of mutations involving the EGFR-mediated signaling can be detected by the analysis of circulating tumor-derived genetic material present as DNA, tumor cells or microvesicles as exosomes [45,149–152]. The liquid biopsy is used to detect resistance mutations against anti-EGFR therapy. However, this analysis is not routinely used to detect the insurgence of resistant tumor cells because the cut-off values for reference detected mutations are not established and there are no well-defined protocols to monitor patients. Anyway, by the use of liquid biopsy the clinician can follow the evolution and the generation of acquired resistance in CRC. Indeed, the treatment with anti-EGFR antibodies can lead to different mutations detectable in signal transduction receptors such as EGFR, HER2 and MET by the use of liquid biopsy [151,152]. It is evident that liquid biopsy can help the clinician in selecting the appropriate therapy but there is not enough evidence to recommend this technique to follow up patients. However, some ongoing clinical trials will reveal whether the liquid biopsy could be an essential tool to identify resistant cells and modulate the therapy consequently (supplemental table I) [153–155].

  • In the manuscript, please, could the authors also list the main trials investigating this setting?

Inserting the word Colorectal cancer and Liquid Biopsy 48 studies have been found in the Clinical Trials.gov site. We then selected the studies where the clinical trials were focused only on CRC or mCRC or CRC II, III, IV grade, and also where the liquid biopsy or circulating tumoral DNA (ctDNA) was used to ameliorate the therapeutic decision by clinician considering the wild type or mutational status of RAS or BRAF or KRAS. Finally, the clinical trials that matched with inclusive criteria are 15 and are listed in the Table 1. This table has been inserted in the manuscript as a supplemental file.

Round 2

Reviewer 1 Report

Comments and Suggestions for Authors

The authors have addressed my comments.